# The Relation between Masticatory Function and Nutrition in Older Individuals, Dependent on Supportive Care for Daily Living

**DOI:** 10.3390/ijerph19105801

**Published:** 2022-05-10

**Authors:** Per Elgestad Stjernfeldt, Gerd Faxén Irving, Inger Wårdh, Robert Lundqvist, Angelika Lantto

**Affiliations:** 1Department of Dental Medicine, Academic Centre for Geriatric Dentistry, Karolinska Institutet, 112 19 Stockholm, Sweden; inger.wardh@ki.se; 2Folktandvården Stockholms län AB, 118 27 Stockholm, Sweden; 3Department of Neurobiology, Division of Clinical Geriatrics, Care Science and Society (NVS), Karolinska Institutet, 141 52 Stockholm, Sweden; gerd.faxen-irving@ki.se; 4Department of Health Sciences, Karlstad University, 651 88 Karlstad, Sweden; 5The Research and Innovation Unit, County of Norrbotten, 971 89 Luleå, Sweden; robert.lundqvist@norrbotten.se; 6The Competence Center of Public Dental Care, County of Norrbotten, 971 28 Luleå, Sweden; angelika.lantto@norrbotten.se

**Keywords:** oral health, older individuals, daily care support, nutrition

## Abstract

Introduction: Associations between masticatory function and nutritional status have been suggested. Masticatory function can be divided into two subdomains, the objective capacity of an individual to mix solid food and the individual’s subjectively assessed ability to masticate solid food. Aim: The aims of this study were to assess the relationship between these subdomains and nutritional variables in older, care-dependent individuals. Materials and methods: From a group of 355 individuals with care dependency and functional limitations, individuals aged 60 and older were selected. By home visits, the subjects underwent an oral examination and answered chewing related questions. Nutritional status was assessed using the Mini Nutritional Assessment. A total of 196 individuals met the age requirement of 60 years or older. Of these, 86 subjects were able to answer the questions. Results: We could not find any concluding significant associations between the subdomains of masticatory function or the nutritional variables. Conclusions: The absence of associations could be explained by the lack of standardized and validated methods to assess masticatory function and they possibly reflect varying underlying constructs. Self-reported questionnaires seem less useful among older and care-dependent individuals, while an objective clinical measurement will be needed when evaluating masticatory function.

## 1. Introduction

One of the primary goals of dentistry is to restore and maintain oral function, especially masticatory function in older individuals. A Swedish study in older, community-dwelling individuals dependent on supportive care for daily living found that the ability to eat, number of teeth and occlusal contacts were vital factors for oral health related quality of life (OHRQoL) [1]. 

Masticatory function is related to food intake (choices of food items) and nutritional status, both important factors for health and well-being in older persons [2]. Malnutrition is related to compromised health and increased mortality [3], and is an important contributor to the etiology of sarcopenia and frailty [4,5]. Researchers have suggested associations between masticatory function and health conditions [6,7,8,9,10], such as nutritional status. A significant relationship between edentulism and intake of key nutrients, such as non-starch polysaccharides, protein, calcium, non-heme iron, niacin, and vitamin C, has been proposed [11]. It has also been suggested that masticatory function could affect body weight [12]. However, a review published in 2002 could only find a weak association between masticatory function and deficient dietary intake [13]. Although subsequently published studies have indicated the existence of such a relationship [14,15,16], a mapping of systematic reviews found no such evidence for nutrition and chewing difficulty [17]. 

Masticatory function is defined as the ability of an individual to masticate solid food [18]. This concept can be further divided into two subdomains. The first subdomain is “masticatory performance” (MP), the objective and quantifiable capacity of an individual to comminute or mix solid food. The other subdomain is “masticatory ability” (MA) and is defined as the individual’s perceived or subjectively assessed ability to masticate solid food. 

These two subdomains—MP and MA—have been shown to correlate weakly or not at all [18,19,20,21], which could indicate that they are not manifestations of the same underlying construct. This would also have implications on their use as measurements for scientific and diagnostics purposes. However, these studies have been conducted in a younger population and not in older individuals with care dependency. Therefore, it would be of interest to assess the relationship between MP and MA in a care-dependent, older population, since these individuals generally have the most compromised dentition and oral health. It would also be important to look for possible relationships between masticatory function and nutritional variables, since malnutrition is common in the care-dependent, older population. This could indicate whether MP or MA should be taken into consideration when assessing nutritional status. 

The aims of the present study were, therefore, to assess the relationship between MP and MA and to assess the relationship between MP and MA and nutritional variables in older, care-dependent individuals.

## 2. Materials and Methods

### 2.1. Study Design and Process

The present study is part of a cross-sectional survey, based on oral examinations and questionnaires among a group of 355 individuals with care dependency and functional limitations. These individuals were randomly selected from the register of increased financial support in Norrbotten County, Sweden, in 2015. The subjects were aged 20 to 97, the majority living in nursing homes or in group housing. They underwent an oral examination and answered chewing related questions. Their cognitive ability to complete these questions was measured using the Short Portable Mental Status Questionnaire (SPMSQ) [22]. For participants with visual or motor impairments, the questions were completed by the study’s research assistant or care personnel/family members. They could be completed at another time and mailed back to the researchers if more time was needed.

The same examiner conducted both the clinical examinations and the other assessments (author AL). 

### 2.2. The Study Group Participants

For the present study, data were collected from the cross-sectional survey described in Section 2.1.

#### 2.2.1. Inclusion Criteria

Only participants 60 years or older were included.

#### 2.2.2. Exclusion Criteria

Individuals who were tube fed were not included in the study.

### 2.3. Ethical Considerations

The original ethical approval was obtained from the Regional Ethical Review Board in Umeå (2013-46-31M). This ethical approval was supplemented to fulfil the present study’s aims and was approved (2019-03620, 21 June 2019). The information about the study was sent to the subjects by letter. They were assured confidentiality and that they could withdraw at any time without negative consequences. When needed, they were assisted by an advocate. Informed consent was given in writing.

### 2.4. Masticatory Performance

MP was assessed by the Eichner index, with and without dentures. The Eichner index is based on occlusal contacts between the premolar and molar regions [23]. These regions are divided into four supporting zones, two in the premolar, and two in the molar regions. Based on the presence or absence of intermaxillary tooth contact in these four zones, a patient is classified as belonging to one of three groups, which are further divided into ten sub-groups. In the present study, we divided the participants into the following three main groups: A, have occlusal contacts in all four posterior support zones; B, have occlusal contacts in one to three zones of contact or within the anterior area only; and C, have no occlusal contacts at all, as shown in Figure 1.

### 2.5. Masticatory Ability

To assess MA, items were selected from the following two questionnaires: a questionnaire from Norrbotten County, named *Oral Health and Bite Function* (OHaBF) [24] and the OHRQoL instrument, *General Oral Health Assessment Instrument* (GOHAI) [25]. 

#### 2.5.1. Items from OHaBF

OHaBF covers different aspects of oral health and masticatory function. In the present study, the following items were used: 

*I need help with my oral health*. The participant can answer yes or no. 

If the answer is yes, one or more of the following items can be chosen:


*Help with pain from the jaws*



*Help to improve my chewing function*



*Help to improve my speaking function*



*Help to improve my appearance*


A further two items are answered on a scale of 1–5, from *very good* to *very poor.*


*Are you able to chew adequately?*



*How would you rate your ability to eat food?*


#### 2.5.2. Items from GOHAI

GOHAI has in its original version 12 items. For the present study, the following items, rated on a scale from 1–5, were used. 


*How often did you limit the kinds or amounts of food you eat because of problems with your teeth or dentures?*



*How often did you have trouble biting or chewing any kinds of food, such as firm meat or apples?*



*How often were you able to swallow comfortably?*



*How often were you able to eat anything without feeling discomfort?*


In the analysis, we dichotomized the answers always/often and never/seldom.

### 2.6. Nutritional Status

Nutritional status was assessed using the Mini Nutritional Assessment (MNA) [26], a commonly used tool for nutritional screening and assessment in older people. For the present study, we used the original/total version with a max score of 30, where 24–30 points indicate normal nutritional status, 17–23.5 at risk of malnutrition and <17 malnutrition. 

Specific items from the MNA tool were selected and analyzed independently. These items were as follows:


*Body mass index (BMI kg/m^2^), which was divided into two groups, BMI < 19–21 and BMI > 21*



*Calf circumference (CC), which was scored as being either less or more than 31 cm*



*Mid-arm circumference (MAC), which was scored as being either less than 21 cm, between 21–22 cm or more than 22 cm*


To explore eating habits, the following items were used: 


*How many main meals do you eat per day?*



*Do you eat two or more servings of fruit or vegetables per day?*



*Can you eat without assistance?*


### 2.7. Statistical Methods 

Data were analyzed using SPSS version 26 (IBM Corp., Armonk, NY, USA). Comparisons between the groups and variables were carried out using Chi-square tests, with the null hypothesis defined as there being no significant relationships between the variables analyzed. A *p*-value of less than 0.05 was considered as significant. 

Each MA variable was analyzed against each MP variable. Each MA and MP variable was also analyzed against each nutritional variable.

## 3. Results

Out of the 355 individuals, a total of 196 individuals met the age requirement of 60 years or older and were included in the present study. Of these, 110 subjects did not have the cognitive ability to answer the MA-related questions. The subjects in this group had fewer occlusal contacts. Eighty-six subjects answered the questions. Even if not statistically significant, the subjects in group Q^0^ were more often assessed as at risk of malnutrition and as malnourished than the subjects in group Q^1^. Moreover, low CC (an indicator of muscle mass) was more common in group Q^0^ and there was a significant difference between Q^1^ and Q^0^ concerning the Eichner index, *p* = 0.03 as well as the total MNA score, *p* = 0.016.

For the study’s participant characteristics, see Table 1.

### 3.1. Masticatory Performance and Masticatory Ability

There was a trend towards a positive relationship between MP and MA. A lower MP was associated with a lower MA, and correspondingly so for higher values, but without any statistically significant relationships. Eighty-eight percent of those in Eichner A group reported that they never/seldom had to limit the kinds or amounts of food they eat because of problems with their teeth or dentures, compared to B (81%) and C (77%) groups. Figure 2 and Figure 3 provide examples of this. Furthermore, no significant statistical relationship was found between the Eichner index with and without dentures, *p* = 0.636 and *p* = 0.903, respectively.

Among those with occlusal contacts in all four posterior support zones (Eichner group A), 93% answered “*never/seldom*” to the question “*How often did you limit the kinds or amounts of food you eat because of problems with your teeth or dentures?*” from GOHAI. In the group with no posterior occlusal contacts at all (Eichner group C), 79% answered “*never/seldom*” to this question. Similarly, 90% of those belonging to group A rated their ability to chew solid food to be “good,” as compared to 77% in group C. 

### 3.2. Masticatory Performance and Nutritional Variables

No significant relationship was found between the Eichner index and total MNA score (*p* = 0.704) (Figure 4) or CC (*p* = 0.810) and MAC (*p =* 0.590). However, a significant relationship was found between the Eichner index without dentures and BMI, X^2^ (N = 191), *p* = 0.015, as shown in Figure 5.

### 3.3. Masticatory Ability and Nutritional Variables

When we looked at the possible relationships between the items from the questionnaires and any of the nutritional items, no significant relationships were found. 

These results, with the exception for the relationship between the Eichner index without dentures and BMI, confirmed the null hypothesis.

## 4. Discussion

In the present study, performed among older vulnerable persons with care-dependency, no associations were found between afunctional dentition and the subjectively perceived ability to chew solid food, or how the individuals subjectively perceived their functional ability. However, less than half of the individuals were able to answer the questions, which is a limitation of the study, as these individuals seemed to be the healthier part of the study material, with more occlusal contacts and higher MNA classification. Among those that answered the questions, the individuals with less occlusal contacts tended to show a lower score on the MA variables but generally, most of these individuals perceived their ability to masticate food to be as good as those who had a fully functional dentition. The majority of those in Eichner group C had, in many cases, used deficient dentures for a longer period, but contrary to what might be expected, this had not resulted in a lower MA. 

The only statistically significant association was between the Eichner index without dentures and BMI. Generally, the participants that belonged to groups B and C had lower BMI scores in comparison with group A, but those in group B had the proportionally lowest BMI scores. They lack occlusal contacts, but perhaps not so many that they have asked for or received tooth replacements. They probably could be defined as frail, suffering from several diseases and functional limitations without adequate support in their daily life, including nutritional needs. 

Furthermore, we did not find any statistical associations between MA and any of the nutritional variables. The individuals with deteriorated masticatory function perhaps do not consider this functional loss, since it is often slowly degraded over a long period of time. An interview study showed that older individuals often overrated their ability to chew food, since they used different coping strategies, such as preparing their meals to make them more easily chewable [27]. Residents at nursing homes with chewing difficulties usually get adapted food consistency. The fact that individuals with compromised MP tend to overestimate their ability to chew food may account for the non-association between MP and MA [28,29]. 

In a study using a two-color chewing gum mixing test as test food, the authors concluded that an association between lower MP and nutritional parameters could not be demonstrated [30]. In another study among older persons in nursing homes [31], where a two-color chewing gum was also used, the absence of teeth and dentures negatively affected oral function but was not associated with the residents’ nutritional status.

We included nutritional-related items from GOHAI as a part of MA. In the original population that served as a base for the present study [32], no association between the Eichner index and OHRQoL could be found. The authors even suggested better OHRQoL for participants with fewer occlusal contacts. Another study found no association between MP and MA after prosthetic rehabilitation [21] and concluded that objective measurement tests are preferable, since MA is more unpredictable. A Japanese study concluded that maintaining at least eight functional tooth units was important in reducing the likelihood of lower MA [33]. Similar results have been showed in other studies [34], but these studies were conducted in younger and healthier adults.

When discussing conflicting results from different studies, it is important to note the context in which the study was conducted, and the study population. It would be erroneous to compare results from studies using healthy, young participants with those using multi-morbid and dependent individuals. Certain items used to assess MA in a young population would not necessarily have the same validity and reliability in an older population [35]. Various authors have concluded that MA decreases with age, despite only minor changes in dental status [36] and despite improved dental state, MA exhibits only minor variation over time [37]. These findings could point to the conclusion that MA encompasses other facets, such as adaptation, eating habits and cultural aspects, that cannot be obtained from objective MP tests. However, it could be discussed whether questions concerning MA are adequate to be used in a population in which a noticeable proportion of individuals are dependent on supportive care for both medical and cognitive reasons, or if it would be better to use MP instead. Furthermore, it is questionable if it is possible to even ask such questions and expect true answers.

Our use of the Eichner index as a measurement of MP could be questioned, but studies have shown that the number of functional tooth units, together with bite force, are the key determinants of MP [38,39]. The Eichner index has not been validated extensively [40], but in the context of the present study where individuals could expect to have varying cooperation possibilities, Eichner index is easily measured at the same time as a dental examination is performed. 

Generally, MP is measured through comminute tests, in which the MP is quantified by assessing how well a test food is masticated into smaller particles [18]. Another method is to assess how well a test food is mixed into a bolus, the so-called mixing test. Other methods have also been developed [38], but none of them are standardized methods or “golden standard” used to assess MP in a clinical context. The lack of established methods to assess MP, or MA, could be one reason why previous studies have yielded inconclusive and conflicting results. 

To the authors’ knowledge, no standardized method has been developed to assess MA. It is usually assessed using questionnaires, some of which are specifically designed to assess MA, while others use a subset of questions from a more generalized questionnaire [18]. None of the items used in the present study have been extensively validated for measuring the construct MA, but instead are used to assess the more generalized concept of “oral health” and OHRQoL.

To measure nutritional status, we used total MNA, as we wanted to include certain items concerning eating habits that are only available in the original form. Otherwise, the short version, MNA-SF, comprising the first six questions of the original MNA, is most often used today [41]. 

MNA-SF is currently the most widely used and recommended tool to assess the risk of malnutrition in older people. BMI in isolation can be misleading, as some older individuals with lower masticatory function adapt by favoring easily chewable [27], or drinkable, high-fat and high-calorie food, which could conceal a lack of important nutrients. However, MNA is still a risk screening tool. According to the Global Leadership Initiative on Malnutrition (GLIM) criteria for malnutrition, at least one phenotype criterion, such as weight loss or low BMI, and one etiologic criterion, such as disease burden and/or decreased food intake, are needed for diagnosis [26,42].

## 5. Conclusions

The absence of association between MP and MA in older, care-dependent individuals could be explained by the lack of standardized and validated methods to assess masticatory function. It is possible that MP and MA do not reflect the same underlying construct. Furthermore, we could only find a significant relationship between MP and BMI, but not with any other nutritional variables. Clinicians and staff working in nursing care also ought to reflect on whether it is adequate to ask the care takers about their ability to eat, both because of cognitive difficulties to answer and since individuals with compromised mastication tend to overrate this ability. Self-reported questionnaires seem less reliable among this vulnerable group, so an objective clinical measurement will be needed when evaluating masticatory function.

## Figures and Tables

**Figure 1 ijerph-19-05801-f001:**
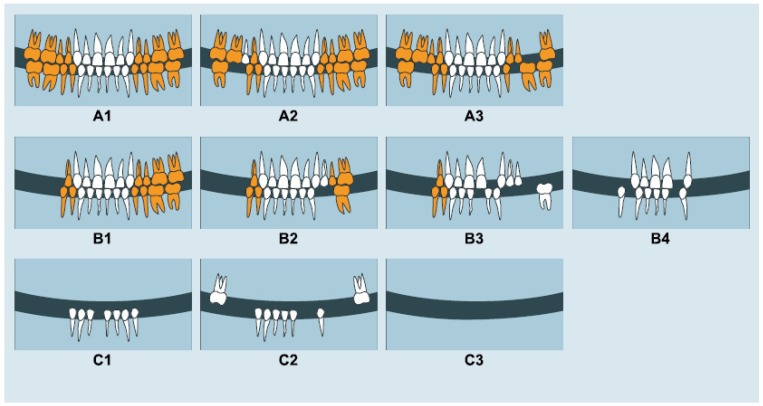
Eichner’s index, where the yellow marked teeth belong to the premolar and molar regions. A have occlusal contacts in all four posterior support zones (**A1**–**A3**); B (have occlusal contacts in one to three zones or within the anterior area only (**B1**–**B4**); and C have no occlusal contacts at all (**C1**–**C3**). The picture was taken with permission from one of the author’s thesis (AL): tooth loss and prosthetic replacements among persons with dependency and functional limitations. https://openarchive.ki.se/xmlui/bitstream/handle/10616/46169/Thesis_Angelika_Lantto.pdf?sequence=1&isAllowed=y (accessed on 23 February 2022).

**Figure 2 ijerph-19-05801-f002:**
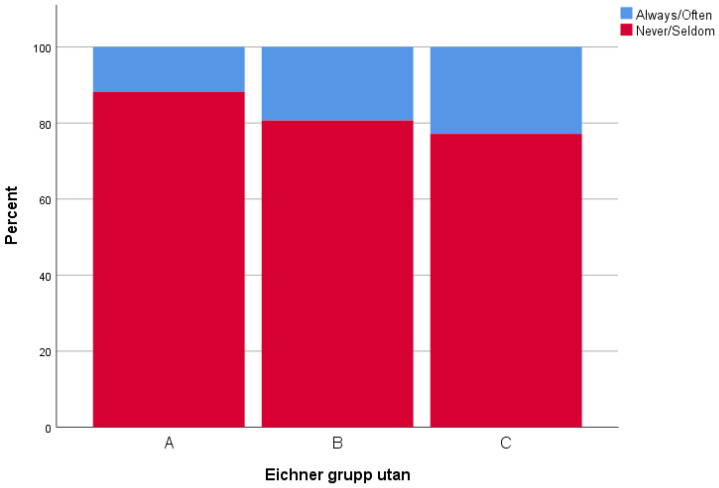
*How often did you limit the kinds or amounts of food you eat because of problems with your teeth or dentures?* Participants grouped according to the Eichner index, which measures number of occluding teeth. (**A**) = four occluding zones, (**B**) = one to three occluding zones or presence of occlusal contacts frontally, and (**C**) = no occluding teeth.

**Figure 3 ijerph-19-05801-f003:**
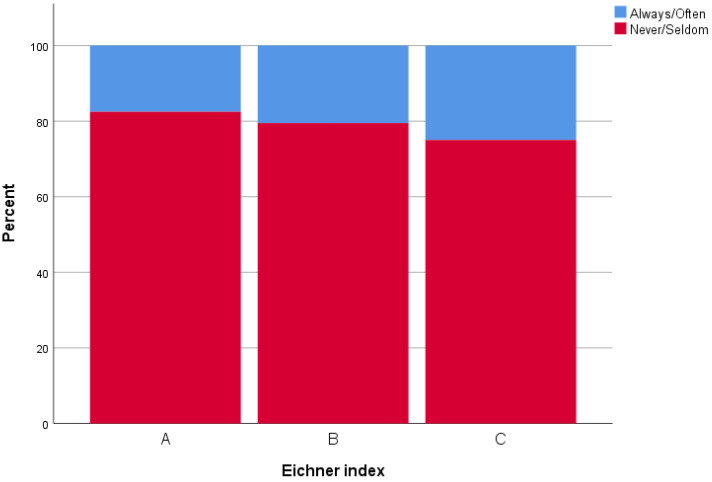
*How often did you limit the kinds or amounts of food you eat because of problems with your teeth or dentures?* Participants grouped according to the Eichner index, while using dentures, which measures number of occluding teeth. (**A**) = four occluding zones, (**B**) = one to three occluding zones or presence of occlusal contacts frontally, and (**C**) = no occluding teeth.

**Figure 4 ijerph-19-05801-f004:**
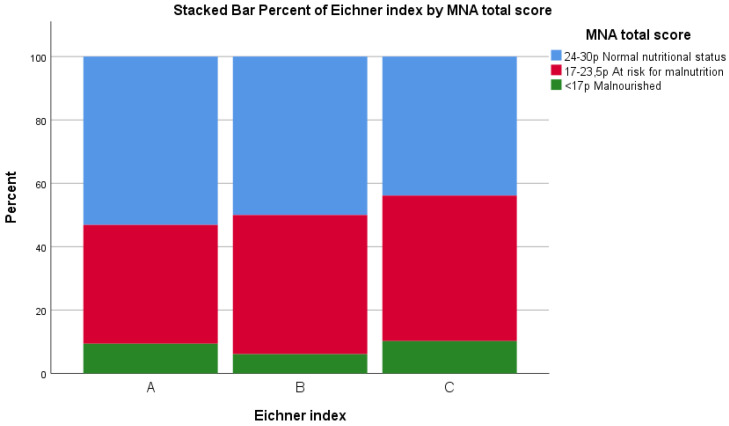
Total individual score of the Mini Nutritional Assessment screening tool. Participants grouped according to the Eichner index, which measures number of occluding teeth. (**A**) = four occluding zones, (**B**) = one to three occluding zones or presence of occlusal contacts frontally, and (**C**) = no occluding teeth.

**Figure 5 ijerph-19-05801-f005:**
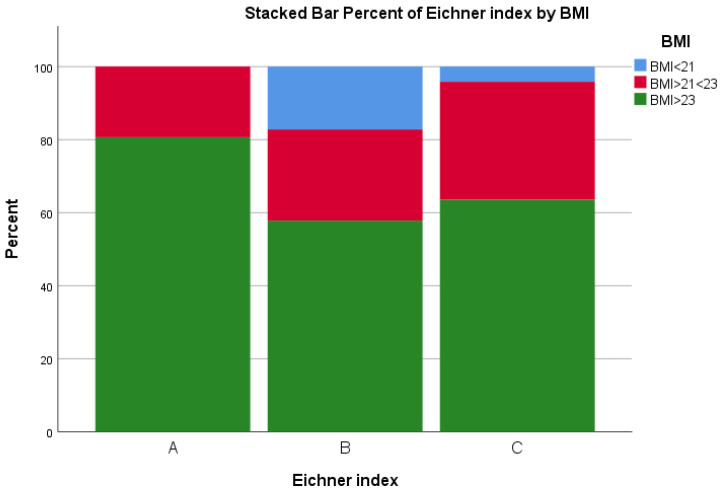
BMI of participants without dentures, grouped according to the Eichner index, which measures number of occluding tooth contacts. (**A**) = four occluding zones, (**B**) = one to three occluding zones or presence of occlusal contacts frontally, and (**C**) = no occluding teeth. *p* = 0.015.

**Table 1 ijerph-19-05801-t001:** Study’s participant characteristics. MNA = Mini Nutritional Assessment. CC = calf circumference. MAC = mid-arm circumference, ^a^ *p* = 0.03, ^b^
*p* = 0.016.

Study’s Participant Characteristics	Total Included n = 196	Answered Questions n = 86 Group Q^1^	Not Answered Questions n = 110 Group Q^0^
Age (yrs) mean ± SD	79.7 ± 10.8	76.7 ± 11.4	81.7 ± 9.8
Female/male, n (%)	119/77 (61/39)	51/35 (59/41)	68/42 (62/38)
Dentures, n (%)	88 (45)	37 (42)	51 (63)
^a^ Eichner index group			
A, n	32	20	22
B, n	66	31	35
C, n	98	35	63
^b^ MNA (0–30 p) median, range	23, 14.5	24, 14	21.6, 13.5
Normal 24–30 p, n (%)	87 (45)	47 (55)	40 (37)
At risk 17–23.5 p, n (%)	88 (45)	34 (40)	54 (49)
Malnourished < 17 p, n (%)	19 (10)	4 (5)	15 (14)
BMI < 21 n (%)	15 (9)	2 (2)	13 (12)
BMI > 21 n (%)	176 (91)	79 (98)	97 (88)
CC < 31 cm, n (%)	35 (18)	8 (9)	27 (25)
CC ≥ 31 cm, n (%)	159 (82)	78 (91)	81 (75)
MAC < 21 cm, n (%)	1 (1)	1 (1)	0 (0)
MAC = 21–22 cm, n (%)	7 (4)	1 (1)	6 (5)
MAC > 22 cm, n (%)	187 (95)	84 (98)	103 (95)
Mode of feeding:			
Need assistance/self-fed with difficulty/self-fed	16/23/157	2/3/83	14/20/74

## Data Availability

The data used in this study belongs to The Research and Innovation Unit, County of Norrbotten, Sweden.

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
