# Peer review of "The Relation between Masticatory Function and Nutrition in Older Individuals, Dependent on Supportive Care for Daily Living"

_ijerph, 2022, doi:10.3390/ijerph19105801_

Round 1

Reviewer 1 Report

Dear Authors,

This paper addresses an interesting topic, however, I would recommend several modifications before considering its publication. Below these are some suggestions for You:

Affiliations: 

  • It is necessary to add them in the manuscript

Abstract: 

  • Line 6: I suggest deleting the comma. 

Material and methods: 

  • Section 2.2. There are others eligibility criteria? 
  • In general, methods are adequately described, but they should be better organized (sections 2.5 and 2.6). 

Results: 

  • table 1. check the row ‘BMI’, the column ‘totally included’ (the is no 100% in total)
  • Figure 2, 3, 4 . Consider adding p-value, even there is no statistical significance 
  • Figure 5. Place the p-value into the caption

Conclusions: 

  • They are supported by the results.

References:

  • In the whole text: the reference numbers should be placed in square brackets []

The following manuscript focuses on the relation between masticatory function and nutrition in older individuals. According to the authors, there is no such study in the older population, and this is the first. The general result is there is no statistical significance between assessed parameters, which may enforce changes in the methods to assess masticatory function. 

areas of strength:

  • the population is not small

areas of weakness:

  • materials and methods as well as results should be correct 

Best regards and good luck

Author Response

Thank you for your valuable comments. We have tried to adress them as described here. 

Affiliations

We have added the author affiliations.

Abstract

We have deleted the comma. 

Materials and methods

2.2 We have revised this section and hopefully clarified the elegibility criteria.

2.5, 2.6 We have revised these section and hopefully structured them in a more adequate way. 

Results

Table 1. Thank you for noticing this. The column is now correct. 

Figure 2, 3, 4 We have addes p-values to each figure. 

Figure 5. P-value has been added to the caption

References

We have added square brackets.

Reviewer 2 Report

Paper’s layout is correct, in line with the guidelines for writing research papers. The methodology section should contain inclusion and exclusion criteria.
Poor quality of graphs - illegible (Figure 2 and Figure 3).
The formatting of Table 1 needs to be improved in some places.
In the description of Table 1, there are statistically significant p-values (p = 0.03 and p = 0.016), but no reference to them or interpretation in the text of the article.
The authors state (164-166) that "There was a trend towards a positive relationship between MP and MA: a lower MP was associated with a lower MA, and correspondingly so for higher values but without any statistical significant relationships."
The article should contain more detailed information on the basis on which the authors reached such a conclusion (statistical data).

Conclusions require elaboration. The aim of the article was: "to assess the relationship between MP and MA and nutritional variables". There is no reference to this objective in the conclusions section. There is only a reference to the relationship between MP and MA.

Author Response

Thank you for your valuable comments. We have tried to adress them as described here. 

1.We have clarified the inclusion and exclusion criteria.

2. We have included new versions of the Figures 1. and 2. The graphs should be of higher quality now. 

3. We have tried to change the format of Table 1.

4. The p-values have been added to the text.

5. We have revised the text including the statement "There was a trend towards a positive relationship between MP and MA: a lower MP was associated with a lower MA, and correspondingly so for higher values but without any statistical significant relationships", and added p-values.

6. We have added   some information in the discussion section concerning the relationship between MP/MA and nutritional variables

Reviewer 3 Report

Dear authors,

Thank you for submitting the current manuscript.

I hope that my suggestions will help you increase the quality of it.

This study investigates the Relation between Masticatory Function and Nutrition in Older Individuals, Dependent on Supportive Care for Daily Living.

Abstract

The abstract is not organized properly. It should be divided into: introduction, aim of the study, materials and methods, discussion, conclusion. Each part of the abstract should be described briefly.

Coclusions in my opinion are presented clearly and well described.

The article is made very precisely and great effort was made in making this research, however more interesting would be to estimate in the future work chewing efficiency in different types of restorations in a laboratory research.

Moderate English corrections should be done.

Best regards

Author Response

Thank you for your valuable comments. We have tried to adress them as described here. 

Abstract

We have addes headings in the abstract.

Round 2

Reviewer 1 Report

Dear Authors,

This is a revised version of your original manuscript. You have corrected it according to the suggestions, but minor changes are required: the abstract should be structured (like now), without headings, and merged.

Best regards and good luck